# Genome Survey and Chromosome-Level Draft Genome Assembly of *Glycine max* var. Dongfudou 3: Insights into Genome Characteristics and Protein Deficiencies

**DOI:** 10.3390/plants12162994

**Published:** 2023-08-19

**Authors:** Yajuan Duan, Yue Li, Jing Zhang, Yongze Song, Yan Jiang, Xiaohong Tong, Yingdong Bi, Shaodong Wang, Sui Wang

**Affiliations:** 1Key Laboratory of Soybean Biology of Chinese Education Ministry, Northeast Agricultural University, 600 Changjiang Road, Harbin 150030, China; duanyajuan2902@163.com (Y.D.); zhly200026@163.com (Y.L.);; 2Institute of Crop Cultivation and Tillage, Heilongjiang Academy of Agricultural Sciences, Harbin 150028, China

**Keywords:** genome survey, next-generation sequencing, *Glycine max*, Dongfudou 3, *k*-mer analysis, organellar DNA, *GmLox* genes

## Abstract

Dongfudou 3 is a highly sought-after soybean variety due to its lack of beany flavor. To support molecular breeding efforts, we conducted a genomic survey using next-generation sequencing. We determined the genome size, complexity, and characteristics of Dongfudou 3. Furthermore, we constructed a chromosome-level draft genome and speculated on the molecular basis of protein deficiency in GmLOX1, GmLOX2, and GmLOX3. These findings set the stage for high-quality genome analysis using third-generation sequencing. The estimated genome size is approximately 1.07 Gb, with repetitive sequences accounting for 72.50%. The genome is homozygous and devoid of microbial contamination. The draft genome consists of 916.00 Mb anchored onto 20 chromosomes, with annotations of 46,446 genes and 77,391 transcripts, achieving Benchmarking Single-Copy Orthologue (BUSCO) completeness of 99.5% for genome completeness and 99.1% for annotation. Deletions and substitutions were identified in the three *GmLox* genes, and they also lack corresponding active proteins. Our proposed approach, involving *k*-mer analysis after filtering out organellar DNA sequences, is applicable to genome surveys of all plant species, allowing for accurate assessments of size and complexity. Moreover, the process of constructing chromosome-level draft genomes using closely related reference genomes offers cost-effective access to valuable information, maximizing data utilization.

## 1. Introduction

Soybean (*Glycine max* (Linn.) Merr.), a member of the family Fabaceae, is a crucial crop that provides a major source of proteins and oils for human consumption and livestock feed worldwide [1]. In addition to its nutritional value, soybean plays a pivotal role in sustainable agriculture by fixing atmospheric nitrogen through a symbiotic relationship with microorganisms [2,3]. Moreover, soybean is also a crop that is highly sensitive to photoperiod and temperature, and there may be significant variations in regional adaptability and yield among different soybean varieties [4,5]. It is generally believed to have originated in China and was domesticated from wild soybean (*Glycine soja* Siebold and Zucc.) [6]. Research on the soybean genome began relatively early. As early as 2010, the soybean cultivar Williams 82’s genome was sequenced and has been consistently updated since then [7]. Over the past decade, several high-quality soybean genomes have been obtained, and a soybean pan-genome has been constructed, ushering in the era of crop pan-genomics [8,9,10]. The abundant genomic information has greatly advanced various studies on soybean. However, compared to the vast number of soybean varieties, the number of varieties with completed whole-genome sequencing remains rare.

With the continuous maturation of third-generation sequencing technologies, particularly PacBio HiFi and ONT ultra-long sequencing, soybean genomes at T2T or even gap-free levels are gradually becoming achievable [11]. However, even with the decrease in the cost of third-generation sequencing, it is still not feasible to sequence thousands of soybean varieties. As a result, next-generation sequencing remains the mainstream approach, with hundreds or thousands of completed sequencing data already stored in public databases. The challenge remains how to better utilize these data. Genome survey sequencing is a technique used to analyze the overall structure of a genome without sequencing it completely. It is often used as a first step in the process of genome sequencing as it provides a broad overview of the genome, such as the genome size, heterozygosity, and the proportion of repetitive sequences, and can help guide subsequent sequencing efforts. Further detailed analysis of these data with a depth of over 50× may provide additional information, including species ploidy, symbiotic microorganisms, historical effective population size, and even a chromosome-level genome assembly. In fact, this already satisfies the needs of a considerable portion of research studies.

Soybean and its products are rich and balanced in nutrients, but a significant portion of the population is sensitive to the “beany flavor” in soybean, which directly reduces their desire to consume soybean. Research shows that lipoxygenases (LOXs) in mature soybean seeds can catalyze the oxidation of unsaturated fatty acids, such as linoleic acid and linolenic acid, producing conjugated unsaturated fatty acid hydroperoxides, which are then converted into volatile compounds, resulting in the beany flavor [12,13]. Dongfudou 3 is a special-purpose soybean variety bred by Northeast Agricultural University and Fumin Seed Group Co. Ltd. (Wudalianchi, China) in Wudalianchi, Heilongjiang Province. The approval number is Heishendou 20190045. The main characteristic of this variety is its lack of “beany flavor”. Protein analysis shows that its seeds do not contain three main types of lipoxygenases: GmLOX1, GmLOX2, and GmLOX3. Due to its excellent characteristics, Dongfudou 3 has been favored by many soybean farmers and downstream processing enterprises. To further understand its genetic features and utilize them in breeding programs, we initiated high-quality whole-genome sequencing. Prior to third-generation sequencing, we conducted a comprehensive genome survey of Dongfudou 3 using next-generation sequencing data. This involved analyzing its genome characteristics, constructing a chromosome-level draft genome assembly, annotating genes, and investigating the molecular mechanism behind the deficiency of LOX proteins. In this study, we attempted to improve the workflow of plant genome *k*-mer analysis by removing the interference of organellar-derived DNA. This approach has broader applicability to genome surveys of various plant species, facilitating more precise assessments of genome size and complexity. At the same time, we also attempted to establish a simple and fast plant genome survey pipeline, which could quickly obtain chromosome-level draft genomes and annotation files, to assist more researchers in obtaining more beneficial information at a lower cost.

## 2. Results

### 2.1. Sequencing and Data Cleaning

To minimize the risk of contamination by endophytic microorganisms, pests, or diseases, strict management practices were implemented throughout the sequencing process. Measures such as seed and substrate sterilization, isolated cultivation, and pest and disease control were employed. After removing adapters and primers, a total of approximately 133.62 Gb of raw data was obtained. Subsequently, trimming was performed to remove low-quality bases at the beginning and end of reads, resulting in clean reads with a length of 140 bp and a total size of 116.28 Gb. The average percentages of Q20 and Q30 were 97.94% and 91.47%, respectively. The filtering rate was approximately 12.98%, and the insert size peak was approximately 260 bp.

### 2.2. Genome Size Estimation by k-mer Analysis

To accurately assess the genome size of Dongfudou 3, it was necessary to remove reads originating from organelles. By carefully controlling the alignment parameters, we aimed to eliminate organelle-derived reads while minimizing the erroneous removal of similar sequences from the nuclear genome. As a result, approximately 11.52% of the reads were discarded, leaving us with 102.88 Gb of clean reads derived from the nuclear genome. Based on the published genome information of several soybean varieties, we estimated the genome size of Dongfudou 3 to be around 1 Gb. Therefore, we set *k* = 17 for *k*-mer analysis. Figure 1 illustrates a main peak at a depth of 83×, while on the right side of the peak, around 165× depth, there is a minor peak formed due to repetitive sequences. Based on the results from GCE, we estimated the genome size of Dongfudou 3 to be approximately 1.07 Gb, with repetitive sequences accounting for approximately 72.50% of the whole genome. Based on these preliminary assessments, the genome of Dongfudou 3 reached a state of near purity, with no significant heterozygous peaks observed.

### 2.3. Preliminary Genome Assembly and GC Content Assessment

In order to obtain higher-quality genome scaffolds, we assessed the optimal assembly *k*-mer. A value of 117 was chosen according to the *k*-mer estimation of KmerGenie. Using SOAPdenovo, we combined short reads to generate a draft assembly, which consisted of 2,196,169 contigs and 2,080,395 scaffolds and covered 1.17 Gb. The overall GC content of the whole contigs was 35.70%. To assess the GC content and sequencing coverage of contigs, and to check for potential contamination such as bacterial sequences, we aligned the clean reads back to the contigs assembled by SOAPdenovo. We selected 49,090 contigs with a minimum length of 5000 bp and calculated their coverage and GC content. As shown in Figure 2, most of the contigs were concentrated in the 30–40% GC content and 70–90× average depth area. In areas over 150× depth, there was also a small number of contigs, which may be repetitive sequences in the genome. We did not observe any clustering of contigs in regions with GC content exceeding 60%, indicating a lack of contamination by bacteria or other organisms in the sample.

### 2.4. Reference-Guided Chromosome Anchoring and Gene Annotation

Thanks to the utilization of the RagTag and Liftoff tools, along with the availability of a high-quality reference genome for soybean, we were able to efficiently perform chromosome-level assembly of the soybean genome. A total of 175,378 scaffolds was successfully anchored onto 20 chromosomes, resulting in a cumulative size of 848.26 Mb, which represents approximately 72.89% of the total genome size. After closing the gaps, the final assembly yielded a chromosome-level genome with a size of 916.00 Mb. The longest contig obtained was 298.19 kb, and the contig N50 was 39.37 kb. We assessed the completeness of our assembly by remapping the BGI short reads, and found that 99.94% of the DNA reads could align properly and the coverage ratio reached 96.04%. The completeness of the assembly was assessed using BUSCO, with a score of 99.5% (S: 43.1%, D: 56.4%, F: 0.3%, and M: 0.2%). Using Liftoff, we mapped the reference genome’s GFF file onto the assembled genome of Dongfudou 3 and performed some corrections to the CDS structures. Initially, we identified 47,794 genes, and after further filtering out genes with incomplete structures, we obtained a final set of 46,446 annotated genes and 77,391 transcripts. We extracted the longest transcript as the primary transcript for each gene and re-evaluated the completeness of the annotated genes using BUSCO. The results showed the Benchmarking Single-Copy Orthologue (BUSCO) completeness was 99.1% (S: 45.7%, D: 53.4%, F: 0.4%, and M: 0.5%). Subsequently, we used the GFF file and the file containing the primary transcripts to extract genes, cDNA, CDS, and protein sequences. For gene functional annotation, we focused on the widely used GO and KEGG databases. In the GO database, a total of 30,797 genes were successfully annotated, accounting for 66.31% of all genes. In the KEGG database, 19,737 genes were annotated, representing 42.49% of all genes. Table 1 presents key statistics for genome assembly and annotation.

### 2.5. Mutation Analysis of Lipoxygenase Deficiency

During the previous variety approval process, Dongfudou 3 was identified as a variety with complete deletions of GmLOX1, GmLOX2, and GmLOX3. Here, we first examined the variations in these three genes at the DNA level. Previous studies have indicated that the three proteins are qualitative traits controlled by two linked genes located on chromosome 13 (*SoyZH13_13G319810*, *GmLox1*; *SoyZH13_13G319800*, *GmLox2*) and one gene located on chromosome 15 (*SoyZH13_15G025600*, *GmLox3*). Based on the principles of collinearity and sequence similarity, we mapped the positions of the allelic genes *GmLox1*, *GmLox2*, and *GmLox3* within the Dongfudou 3 genome and examined their potential variations. Specifically, we initially identified the genomic locations of these genes as follows: *GmLox1* on chromosome 13, spanning from 40,034,891 to 40,038,646 bp; *GmLox2* on chromosome 13, spanning from 40,027,569 to 40,031,512 bp; and *GmLox3* on chromosome 15, spanning from 2,052,600 to 2,056,693 bp. During subsequent gene structure annotation, we focused on the variations observed in the coding sequences (CDSs) of the three genes. As shown in Figure 3, compared to the corresponding alleles in Zhonghuang 13, *GmLox1* in Dongfudou 3 exhibited three mutations: a T to A mutation at position 114, which did not alter the amino acid residue; a C to A mutation at position 157, resulting in the substitution of asparagine with histidine; and a fragment deletion from position 1575 to 1648. *GmLox2* had one mutation: a T to A mutation at position 1596, leading to the substitution of histidine with glutamine. *GmLox3* exhibited one mutation: a deletion of a G base at position 101, resulting in premature termination of the encoded protein. These variations were also confirmed by Sanger sequencing (Figure 4).

### 2.6. Impact of GmLox Genes Variations on Transcription and Translation Processes

To verify the expression status of these three mutated genes in Dongfudou 3 at the RNA level, we conducted qRT-PCR analysis. The results in Figure 5 demonstrate that compared to the control Zhonghuang 13, the expression levels of *GmLox1* and *GmLox3* in Dongfudou 3 are close to negligible, while the expression level of *GmLox2* is only approximately 1/8 of the control. These findings indicate that these genes in Dongfudou 3 exhibit either near-zero expression or extremely low expression levels at the RNA level.

To determine the presence or absence of the three lipoxygenase enzymes (GmLOX1, GmLOX2, and GmLOX3) in Dongfudou 3 seeds at the protein level, we employed a colorimetric assay. As controls, we included the non-deficient variety Zhonghuang 13, which is known to possess three lipoxygenase enzymes. Figure 6 shows that in comparison to the controls, Dongfudou 3 maintains the color of the substrate solution in all treatments, indicating the absence of GmLOX1 (solution remains blue), GmLOX2 (solution remains blue), and GmLOX3 (solution remains yellow). These results align with the SDS-PAGE gel electrophoresis conducted during the variety assessment of Dongfudou 3, providing further confirmation of the protein-level deficiency of GmLOX1, GmLOX2, and GmLOX3.

To further analyze whether a single nucleotide variation in the *GmLox2* gene CDS of Dongfudou 3 affects its protein structure, we utilized AlphaFold2 to predict and compare the GmLOX2 proteins from Zhonghuang 13 and Dongfudou 3. Figure 7 illustrates the structural representation of the two proteins overall and in the vicinity of the mutation site. The results reveal that the substitution of asparagine for histidine at position 532 of the GmLOX2 protein does not alter the overall spatial structure of the protein. However, it leads to the absence of an imidazole group at the mutation site.

## 3. Discussion

The estimation of genome size is a crucial task in genome surveys. Determining the genome size is of significant importance for selecting sequencing strategies, sequencing depth, and evaluating assembly outcomes. Initially, flow cytometry was predominantly used to assess genome size, but the results varied widely, making it challenging to establish a unified standard [14,15]. Therefore, it is currently employed as a rough estimation method for genome size. Currently, the most prevalent method for genome size estimation is primarily based on *k*-mer analysis [16]. This approach is widely regarded as highly accurate and enables the assessment of genome heterozygosity and repetitive sequence content. As a result, it has been extensively applied and has given rise to various evaluation software tools. Currently, this strategy has demonstrated relatively accurate performance in genome size estimation for the majority of microorganisms and animals. However, its performance in plant genome surveys has been less satisfactory. In practice, there are often significant differences in the results obtained from sequencing different individuals or different tissue types, or when using different evaluation software. These discrepancies are difficult to explain solely based on measurement errors or other factors. This discrepancy may be attributed to the fact that plant cells generally contain a larger number of organelles, such as mitochondria and plastids, the genomes of which are significantly larger than those of organelles in animal cells. Due to the current use of sequencing data that include the entire DNA content of plant cells in most plant genome surveys, the inevitable presence of organelle DNA can affect the total count and distribution of *k*-mers, thereby leading to inaccurate assessment results. Additionally, the variation in the number of organelles in plant cells among different individuals, tissues, or growth stages further contributes to varying proportions of organelle-derived DNA in sequencing results, which can have a significant impact on the final outcomes [17]. In fact, a few researchers have already recognized this issue and incorporated strategies to remove high-frequency *k*-mers that may originate from organelle sequences into relevant software tools [18]. However, it is evident that this strategy of filtering out organelle sequences can only serve as a temporary solution. High-frequency *k*-mers derived from nuclear sources, such as telomeres, centromeres, or other repetitive regions, are often inadvertently filtered out as well. Moreover, if plant samples are contaminated with microorganisms or harbor endophytic bacteria, the indiscriminate removal of high-frequency *k*-mers may compromise the subsequent assessment of sample purity. In this study, we employed a direct mapping approach, aligning reads to a reference genome of soybean organelles, to filter out sequences originating from organelles. This approach is primarily based on the highly conserved nature of plant organelles. On the one hand, plastid genomes exhibit a high degree of sequence and structural conservation across different species [19]. On the other hand, although plant mitochondrial genomes may display higher recombination rates among different varieties or individuals, their overall mutation rate is extremely low, resulting in a high level of sequence conservation [20]. Therefore, the impact of this filtering method on the accurate retrieval of most short reads is minimal. In this study, we selected the plastid and mitochondrial genomes of the Williams 82 soybean as references. There is an important consideration to note here. During the course of extensive evolution, certain sequences from organelle genomes originating from endosymbiotic events have been transferred to the nucleus [21,22]. Although accumulated mutations have resulted in sequence differences between the two, to minimize the excessive removal of similar sequences from the nucleus during the alignment process, we also controlled the alignment parameters and modified the scoring matrix accordingly. Considering the error rate of next-generation sequencing and the sequence variations among different varieties, we made several attempts and examined the BAM files. Eventually, we allowed a maximum of three base mismatches or two base deletions, or one base deletion combined with one base mismatch in each aligned read. In fact, during the examination of the BAM files, we focused on the detection of heterozygous variant sites and coverage. Since most angiosperm organelles are maternally inherited, the likelihood of multiple variants occurring at the same position in a single sample is extremely low. Therefore, an excessive number of detected heterozygous sites after alignment is likely due to mismatches. Additionally, the conservation of organelle genomes among closely related species means that low coverage may indicate overly stringent alignment parameters or significant differences between the selected organelle reference genome and the sample. In such cases, parameter adjustments or the selection of a more suitable organelle reference genome are necessary. In order to better determine the sequence differences between the organelle genomes of Dongfudou 3 and Williams 82, we also attempted a preliminary assembly of the organelle genome of Dongfudou 3 using GetOrganelle. We compared it roughly with the sequence of Williams 82 to evaluate our parameter settings and scoring matrix. This approach can also be applied to samples without closely related reference organelle genomes or samples with significant differences from the reference organelle genome. During the assembly process, we did not concern ourselves with the completeness of the organelle genome. In fact, the results of assembly software, such as GetOrganelle, typically include most of the organelle genome, which is usually sufficient for filtering out organelle-derived DNA. The impact of a small amount of remaining or excessively filtered sequences on the results is minimal and can be ignored. We also attempted an evaluation without removing the organelle DNA. With all other parameters unchanged, the resulting size was approximately 1.21 Gb, which is about 140 Mb larger than the result obtained after removing organelle DNA. This size difference is significantly larger than several published soybean genomes and deviates from the general understanding. However, after filtering out organelle-derived DNA, our results were more consistent with the recently assembled high-quality soybean genome using third-generation sequencing, demonstrating the effectiveness and accuracy of our strategy for estimating plant genome size.

GC content evaluation is also an important component of plant genome surveys. It involves plotting the GC content and coverage distribution of preliminary assembled contigs to determine if there are multiple enrichment centers, which can indicate heterozygosity and contamination levels. Although not absolute, there is generally a difference in GC content between plant genomes and microbial genomes [23,24]. Researchers often use scatter plots on the GC content axis to preliminarily assess the presence of microbial contamination in plant samples. The existence of multiple enrichment centers on the GC content axis can indicate the potential presence of microorganisms. Additionally, highly heterozygous regions of the genome are often assembled, which can result in lower coverage during the alignment of reads. Researchers also use scatter plots on the coverage axis to preliminarily assess the homogeneity of the sample genome based on the presence of multiple enrichment centers. Indeed, we had concerns about the heterozygosity of the selected Dongfudou 3 sample prior to sequencing. Despite soybean being a self-pollinating crop, the creation of new varieties often involves hybridization. Dongfudou 3 is a variety obtained through hybridization, and although it has undergone ten consecutive generations of self-pollination, theoretically approaching homozygosity, we still had doubts. Therefore, we conducted an analysis of its heterozygosity using *k*-mer analysis and GC content distribution to gain more insights. Another concern of ours was bacterial contamination, particularly from endophytic bacteria. During the genome assembly of previous species, we often encountered cases where samples were contaminated with endophytic bacteria, leading to significant issues during subsequent assembly and annotation processes [25]. Given the symbiotic nitrogen fixation and other characteristics of soybean, the likelihood of microbial presence, including endophytic bacteria, in its tissues is relatively high. Some preliminary studies have also confirmed the abundance of symbiotic microorganisms in soybean [26,27]. While we took certain disinfection measures during the experiments, the entire process was conducted in an open environment, making it difficult to completely avoid the introduction of microorganisms. However, it was crucial to ensure that their presence in the DNA was minimal and would not significantly impact the genome survey. According to the results of this study, the GC content and coverage of contigs assembled from the Dongfudou 3 sample were concentrated around a single center, indicating a low level of genome heterozygosity and no apparent contaminants. This finding demonstrates that the sample fully meets the requirements for subsequent deep third-generation sequencing.

In this study, we also attempted to construct chromosome-level genome scaffolds and annotate gene structures and functions using survey sequencing data. This clearly expanded the application scope of the data and improved their utility. Clearly, this low-cost sequencing and assembly approach may not suffice for the detection and analysis of repeats, heterozygosity, or complex regions. However, the BUSCO assessment results indicate that the majority of the gene regions have been well assembled and annotated. This approach may be particularly effective for species or varieties that lack genome information but have high-quality reference genomes from closely related species. It is especially suitable for experiments that do not require a high level of chromosome-level genome assembly, such as gene cloning, gene editing, transcriptomics, SNP or indel variant mining, and other studies with lower demands for chromosome-level genome resolution. This simple and cost-effective approach for draft genome construction provides a convenient solution for researchers working with understudied species, those with lower requirements for genome quality, or those facing budget constraints.

Dongfudou 3, a commercially available non-beany soybean cultivar, has gained wide market acceptance. However, the molecular basis underlying the absence of proteins, such as GmLOX1, GmLOX2, and GmLOX3, in this cultivar has remained unclear for a long time due to the lack of genomic information. Here, based on the survey genome draft, we aimed to preliminarily speculate the molecular mechanisms underlying the absence of lipoxygenase enzymes and demonstrate that such assembly results can fulfill the requirements for detecting small fragment variations in most gene regions. Leveraging collinearity and sequence similarity, we conveniently mapped the positions of the *GmLox1*, *GmLox2*, and *GmLox3* genes in the Dongfudou 3 genome and identified their variation sites. The reliability of the sequencing assembly was also confirmed through Sanger sequencing results. Additionally, we employed fluorescent quantitative PCR to detect that the expression levels of GmLox mutant genes at the RNA level were reduced to nearly zero or only about 1/8 of the control. We also utilized colorimetric assays to detect the absence or inactivation of these three proteins at the protein level.

Clearly, it is easy to understand that the DNA-level base deletions in *GmLox1* and *GmLox3* lead to incomplete CDS structures. Concurrently, the expression of the two genes was also close to zero at the RNA level, and protein-level detection indicated the absence of these two proteins. The nucleotide substitution in the *GmLox2* gene, however, did not alter the CDS structure. Although its expression is significantly reduced at the RNA level, we still harbor concerns regarding the potential function of the protein it may translate into. Therefore, we focused on whether a single nucleotide mutation in the *GmLox2* gene affects protein activity. Protein structure prediction revealed that the overall structure of the GmLOX2 protein remained largely unchanged before and after the mutation. However, the mutation caused the amino acid residue to change from histidine to asparagine, resulting in the loss of an imidazole moiety at the mutation site. Previous studies have underscored the critical importance of histidine residues in protein–metal ion interactions. The imidazole side chain of histidine often serves as a ligand in metalloproteins, engaging in coordination bonds with metal ions. This is facilitated by the imidazole side chain’s pair of nitrogen atoms, with one acting as a positively charged ligand. Histidine’s ability to bind metal ions, such as iron, copper, and zinc, enables its involvement in a range of essential biological functions [28]. Among these, six histidine residues are conserved across all lipoxygenase sequences and are considered potential iron ligands [29]. Further investigations on soybean GmLOX1 and GmLOX3 proteins have provided additional evidence for the crucial role of these six histidines in maintaining Fe binding and protein activity [30,31]. Indeed, preliminary analysis of soybean varieties with a deletion in the GmLOX2 protein revealed that the substitution of histidine with asparagine at position 532 in GmLOX2, aligned with histidine at position 504 in GmLOX1, is a common occurrence. It is likely to disrupt the binding of the protein to iron, leading to the loss of GmLOX2 activity [32]. However, it is evident that we cannot definitively confirm, at this stage, that these mutations are responsible for the deficiency or inactivation of these three proteins. This is because gene transcription and translation involve multiple processes influenced by complex factors, including, but not limited to, variations in the coding region, promoter region, upstream transcription factors, and various epigenetic regulations. Another point of interest for us is the significant reduction in the expression level of the *GmLox2* gene in Dongfudou 3. Given its close linkage with *GmLox1*, we investigated whether mutations in *GmLox1* could potentially impact the transcription of *GmLox2*. The preliminary assembly of the genome only assists us in preliminarily identifying gene variations; the accurate determination of their causes will rely on further molecular experiments for validation in the future.

In summary, this study utilized deep second-generation sequencing to perform a genome survey analysis of the soybean variety Dongfudou 3. It accurately assessed its genome size, complexity, and contamination status. A particular focus was placed on optimizing the *k*-mer analysis process and filtering out potential influences from organelle DNA sequences on genome size estimation. This strategy is applicable not only to soybeans but also to all plant species. It significantly reduces the risk of over-assembly or collapse in the later stages of genome assembly caused by erroneous genome size estimation. However, it should be noted that the reference genome and alignment parameters used in this study may not be optimal for other species and would require further optimization and adjustment. Furthermore, the study also attempted to construct chromosome-level draft genomes using only deep second-generation sequencing data combined with information from closely related reference genomes. The exploration of the molecular mechanisms behind the loss of three GmLOX proteins demonstrated that the draft genomes obtained through this strategy could meet the needs of a substantial portion of research and have practical value. This strategy is also applicable to other species, with the assembly quality highly dependent on the quality of the selected reference genome and its phylogenetic relationship with the study species. In the future, this information will be used to design and optimize sequencing and assembly protocols for the Dongfudou 3 genome, aiming to obtain a high-quality genome.

## 4. Materials and Methods

### 4.1. Plant Materials and Sequencing

Healthy and plump seeds of the self-pollinated 10th generation of Dongfudou 3 were surface sterilized with 75% ethanol for 30 s, washed 3 times with sterile water, and soaked in the dark for 4 h. The soaked seeds were placed in a Petri dish with moistened filter paper and germinated in the dark at 24 °C for 5 d, then transferred to a hydroponic tank and grown in 1/2 Hoagland nutrient solution at 24 °C, 12 h light/12 h dark cycle. When the plantlets reached the V3 stage, a fully expanded trifoliate leaf was taken and DNA was extracted using a genomic DNA kit (DP3111, Biotake Corporation, Beijing, China). The DNA was transported to Frasergen Bioinformatics Co. Ltd. (Wuhan, China) on dry ice for library construction and sequencing. A paired-end library with reads of 150 bp and an average insert size of ~300 bp was constructed and sequenced on the MGISEQ-2000RS platform (BGI, Shenzhen, China). Figure 8 illustrates the morphological characteristics of the plants and seeds of Dongfudou 3.

### 4.2. Quality Control and Data Cleaning

To obtain high-quality and vector/adaptor-free reads, raw reads were filtered using fastp (v.0.23.2, main options: -f 8, -t 2, -n 0, -l 140, -q 20, -u 20), with quality checks by FastQC (v.0.11.9) before and after filtering.

### 4.3. Genome Size and Repeat Rate Estimation by k-mer Analysis

To remove the sequences derived from organelles in the DNA, we downloaded the plastid (MZ964145.1) and mitochondrial (MW331583.1 and MW331584.1) genomes of soybean Williams 82 from NCBI as references. We then used Bowtie2 (v.2.4.4) to align the clean reads to the reference sequences. We strictly controlled the alignment parameters and allowed, at most, one alignment with no more than 3 mismatches or 2 gaps, or 1 gap plus 1 mismatch. Then, samtools (v.1.15.1) was used to convert the format, and bedtools (v.2.29.2) was used to extract the unmapped fastq files, which only came from the nuclear genome of soybean. Finally, genome size, repeat rate, and heterozygous rate were calculated using KmerFreq (v.4.0) and GCE (v.1.0.2) based on a 17-mer distribution [16]. The *k*-mer distribution was drawn using Excel 2019.

### 4.4. Preliminary Genome Assembly and GC Content Analysis

Before assembly, KmerGenie (v.1.7051) was used to select optimal *k*-mer sizes for assembly [33]. Then, SOAPdenovo2 (v.2.04) was used to build the initial contigs and scaffolds using the selected optimal *k*-mer. Next, we remapped the clean reads to the assembled contigs using bwa-mem2 (v.2.0pre2) and then converted the SAM file to BAM file using samtools. Subsequently, we utilized a self-written script to extract the GC content and coverage depth of contigs longer than 5000 bp. Finally, the densCols function in R (v.4.1.3) was used to plot the GC depth distribution of the contigs.

### 4.5. Fast Reference-Guided Chromosome Anchoring and Gene Annotation

Draft scaffolds were ordered and oriented by aligning them to the reference genome of *Glycine max* var. Williams 82-ISU-01 (v.2.0, downloaded from Phytozome V13 website) using RagTag v2.1.0 to generate a chromosome-level draft genome [34]. Subsequently, we removed the fragments that did not successfully anchor to the chromosomes and applied GapCloser (v.1.12) to fill the gaps, resulting in the final chromosome-level genome assembly. Next, the gff3 file of the reference genome Williams 82-ISU-01 was transferred into our new draft genome utilizing Liftoff (v.1.6.3) to perform gene annotation with default parameters [35]. We further filtered the results by protein integrity and obtained the final gff3 annotation file. Subsequently, we extracted the longest transcript to generate the primary-transcript-only gff3 file and used the two gff3 files to generate corresponding gene, cDNA, CDS, and protein fasta sequence files.

For gene function annotation, we only performed the primary GO and KEGG annotations on the predicted genes. GO terms were assigned using Interproscan (v.5.54-87.0) and eggNOG database (v.6.0, http://eggnog6.embl.de/, accessed on 12 April 2023). We integrated the two annotation results and filtered them based on go-basic.obo (v.1.2, released on1 April 2023, downloaded from http://geneontology.org/, accessed on 12 April 2023) to ensure that the results were acyclic and non-redundant. KEGG ortholog (KO) annotation was performed using KofamKOALA (v.1.3.0, released on 1 April 2023) [36].

To assess the quality of genome assembly and annotation, QUAST (v.5.2.0) was used to calculate some basic statistics such as the number of contigs, contig N50, and GC content [37]. Then, bwa-mem2 (v2.0pre2) was utilized to remap the DNA short reads to the draft genome; then, the mapping rate and coverage ratio were calculated with samtools to assess the correctness and completeness of the draft genome. In addition, BUSCO (v.5.3.2) analysis was used to assess the completeness, redundancy, and accuracy of the draft genome and the predicted genes with the embryophyte_odb10 dataset, which contains 1614 Embryophyta single-copy orthologs [38].

### 4.6. Sequence Variations of Seed Lipoxygenases and Their Detection at the Transcriptional and Translational Levels

At the DNA level, we used the modified *GmLox1*, *GmLox2*, and *GmLox3* gene sequences from *Glycine max* var. Zhonghuang 13 to locate the corresponding alleles on the Dongfudou 3 genome, utilizing blastn (v.2.12.0+) and chromosomal collinearity information [39]. Sequences were aligned using MUSCLE in BioEdit (v.7.2.5), followed by manual identification of the variation sites. We also designed primers (Table 2) based on the variant site information for PCR amplification, and sent the products to Shanghai Sangon Biotechnology Co. Ltd. (Shanghai, China) for Sanger sequencing to confirm the accuracy of variant detection. At the transcriptional level, total RNA was extracted from Zhonghuang 13 and Dongfudou 3 seeds using a Plant RNA Extraction Kit (RP3301, BioTeke, China), followed by reverse transcription into cDNA using a ReverTra Ace^®^ qPCR RT Kit (FSQ-101, Toyobo, Japan). *GmCYP2* (*SoyZH13_12G024400*) served as the internal reference, and quantitative PCR primers were designed for the three *GmLox* genes in non-mutated regions. These primers were also synthesized by Shanghai Sangon Biotechnology Co. Ltd (Shanghai, China). And are listed in Table 2. Three technical replicate qRT-PCRs were performed for each biological replicate using SYBR Green Realtime PCR Master Mix (QPK-201, Toyobo, China) in an AriaMx real-time PCR system (Agilent Technologies, Santa Clara, CA, USA). The amplification conditions were as follows: pre-denaturation at 95 °C for 1 min, denaturation at 95 °C for 15 s, 45 cycles, extension at 60 °C for 1 min, with Rox as the reference dye. After the last polymerase chain reaction cycle, the melting curve was generated under the conditions of 95 °C for 15 s, 65 °C for 1 min, and 95 °C for 15 s. The $2^{{-\Delta \Delta C}_{T}}$ method was employed to calculate the gene expression data [40], with Zhonghuang 13 genes used as the control. At the protein level, as previously described, we used a colorimetric assay method to measure the activity of lipoxygenases GmLOX1, GmLOX2, and GmLOX3 in soybean seeds [13,41,42]. Finally, AlphaFold (v.2.3.2) was used to predict the potential impact of point mutations in GmLOX2 on protein structure [43]. Visualization and comparison of protein structures were performed using PyMOL (v.2.5).

## 5. Conclusions

In this study, we performed a genome survey analysis of the soybean variety Dongfudou 3 using next-generation sequencing, accurately assessing its genome size and complexity. The estimated genome size of Dongfudou 3 was approximately 1.07 Gb, with repetitive sequences accounting for about 72.50% of the genome. The sample appeared to be nearly homozygous, and no significant microbial contamination was detected. Leveraging the soybean reference genome, we assembled a chromosome-level draft genome of Dongfudou 3, with a total of 916.00 Mb sequences anchored onto 20 chromosomes. Through genome annotation, we identified 46,446 genes and 77,391 transcripts. The genome and annotation completeness were evaluated using BUSCO, with assessment scores of 99.5% and 99.1%, respectively. Based on the draft genome, we also conducted a preliminary analysis of the molecular mechanisms underlying the protein loss of GmLOX1, GmLOX2, and GmLOX3 in Dongfudou 3. We identified several base deletions and frameshift mutations in the coding regions of *GmLox1* and *GmLox3*, which may lead to protein inactivation. Additionally, a single-base mutation in the coding region of *GmLox2* resulted in the substitution of glutamine with histidine in the iron-binding active site, which may lead to loss of enzymatic activity. This study not only helps us design further strategies for constructing a high-quality genome of Dongfudou 3 using third-generation sequencing, but also provides a preliminary understanding of certain characteristics of Dongfudou 3 through the assembled draft genome. Moreover, this research has improved the accuracy of estimating plant genome size based on *k*-mer analysis and tested the process of constructing draft genomes using next-generation data and closely related reference genomes. These findings serve as a valuable reference for conducting plant genome survey analyses on other species.

## Figures and Tables

**Figure 1 plants-12-02994-f001:**
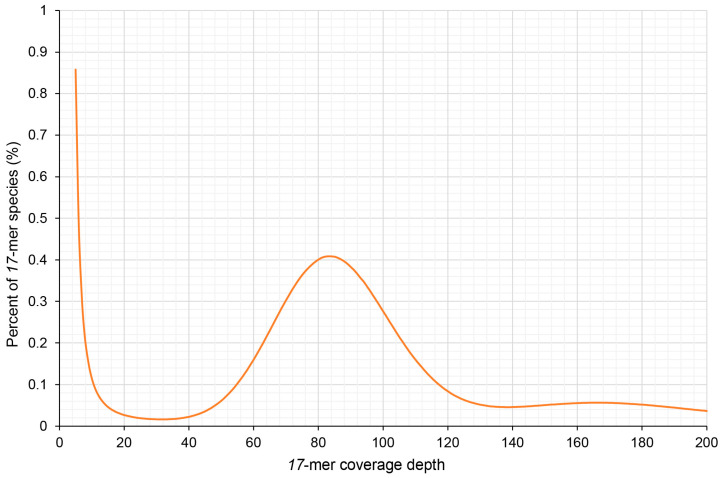
Distribution of *17*-mer frequency of *Glycine max* var. Dongfudou 3.

**Figure 2 plants-12-02994-f002:**
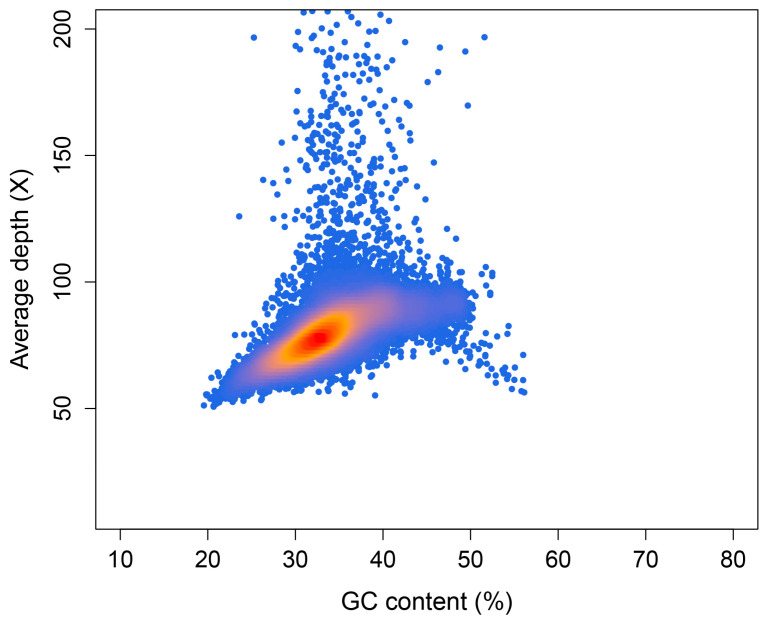
GC depth distribution of the *Glycine max* var. Dongfudou 3 genome. The X-axis is the GC content, and the Y-axis is the average depth. Each dot represents a contig. Color from red to orange and then to blue indicates dot density from high to low.

**Figure 3 plants-12-02994-f003:**
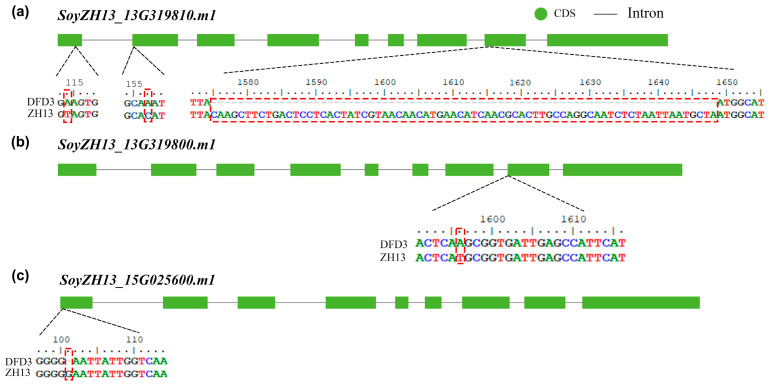
Schematic representation of sequence variations in the *GmLox1*, *GmLox2*, and *GmLox3* genes in Dongfudou 3. The structural diagram represents the corrected structure of the longest transcript corresponding to *GmLox1*, *GmLox2*, and *GmLox3* in Zhonghuang 13. The positions of variations between Zhonghuang 13 and Dongfudou 3 are highlighted in red dashed boxes. The labels “DFD3” and “ZH13” represent Dongfudou 3 and Zhonghuang 13, respectively. Please note that the coordinates in the figure are based on the CDS file of the modified Zhonghuang 13 reference gene, where the first base of the start codon is designated as coordinate 1. The coordinates do not include intronic or other non-coding sequences. The coordinates in the figure are depicted in a 5’ to 3’ orientation, progressing from left to right. (**a**) Schematic representation of the variant sites in *GmLox1*. (**b**) Schematic representation of the variant sites in *GmLox2*. (**c**) Schematic representation of the variant sites in *GmLox3*.

**Figure 4 plants-12-02994-f004:**
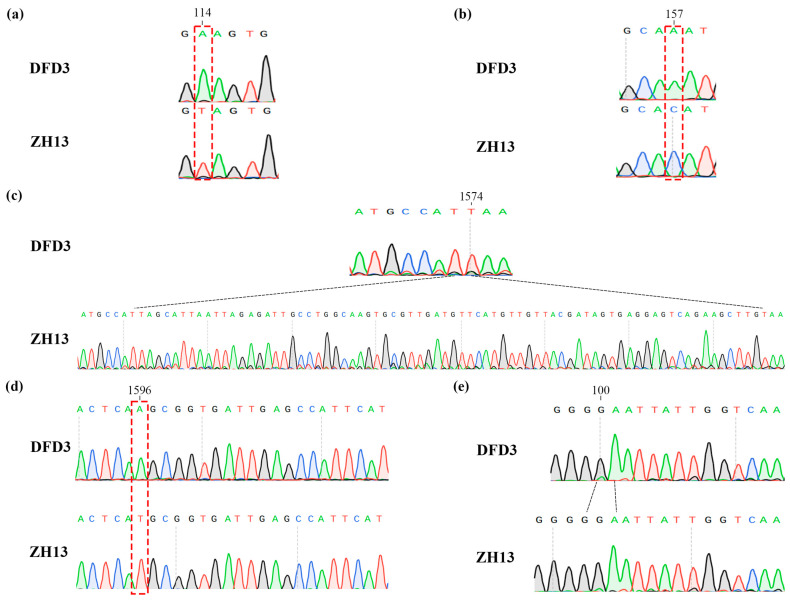
Validation of mutations in the *GmLox1*, *GmLox2*, and *GmLox3* genes in Dongfudou 3 using Sanger sequencing. The labels “DFD3” and “ZH13” represent Dongfudou 3 and Zhonghuang 13, respectively. Mutations are indicated by red and black dashed lines at their respective positions. (**a**) Schematic representation of the mutation at position 114 in the CDS of *GmLox1*. (**b**) Schematic representation of the mutation at position 157 in the CDS of *GmLox1*. (**c**) Schematic representation of the base deletion in the CDS of *GmLox1*. (**d**) Schematic representation of the mutation at position 1596 in the CDS of *GmLox2*. (**e**) Schematic representation of the base deletion in the CDS of *GmLox3*.

**Figure 5 plants-12-02994-f005:**
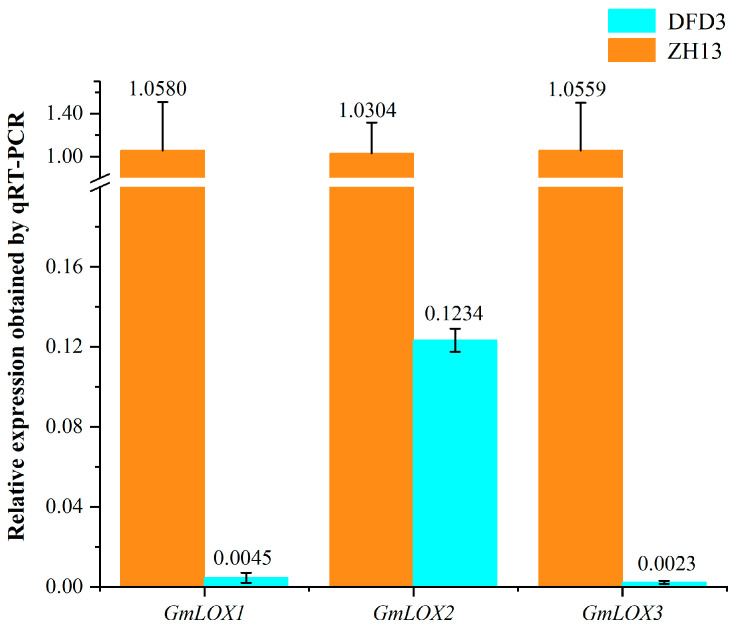
qRT-PCR measurement of 3 *GmLox* genes in Zhonghuang 13 and Dongfudou 3 seeds. In the experiments, we assumed an amplification efficiency (E) of 100%, and each sample was subjected to 3 biological replicates. The internal reference gene chosen was *GmCYP2* (*SoyZH13_12G024400.m2*). The labels “DFD3” and “ZH13” represent Dongfudou 3 and Zhonghuang 13, respectively.

**Figure 6 plants-12-02994-f006:**
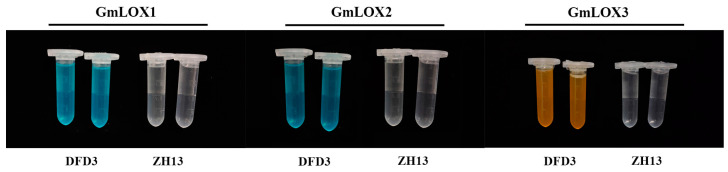
Determination of soybean GmLOX1, GmLOX2, and GmLOX3 activities using a colorimetric assay. The labels “DFD3” and “ZH13” represent Dongfudou 3 and Zhonghuang 13, respectively.

**Figure 7 plants-12-02994-f007:**
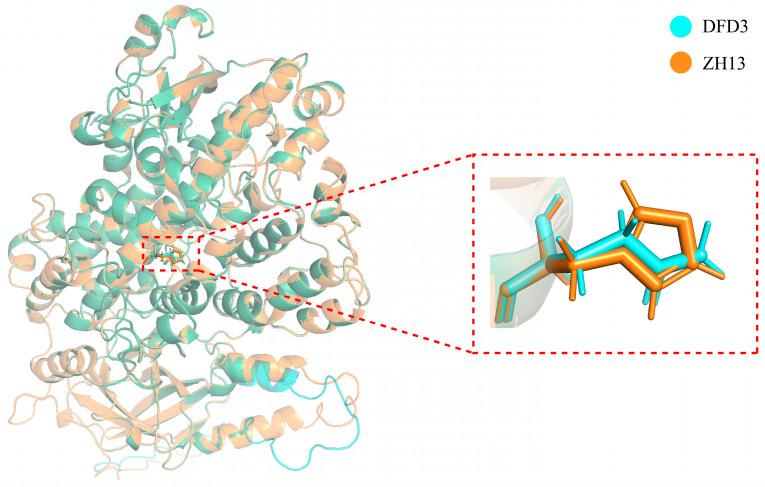
Structural representation of soybean LOX2 protein and illustration of the mutation site.

**Figure 8 plants-12-02994-f008:**
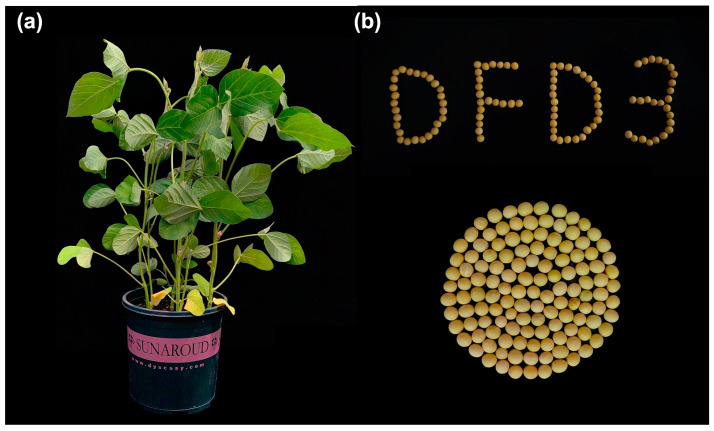
The morphological characteristics of *Glycine max* var. Dongfudou 3. (**a**) The plant. (**b**) The seeds.

**Table 1 plants-12-02994-t001:** Statistics for the genome assembly and annotation.

Item	Value
Assembled genome size (Mb)	916.00
Number of contigs	60,936
Number of Scaffolds	20
N50 of contigs (bp)	39,370
N90 of contigs (bp)	9078
Longest contig (bp)	298,193
GC content (%)	34.3
BUSCO complete of the genome (%)	99.5
BUSCO complete of the genes (%)	99.1
Number of protein-coding genes	46,446
Number of genes annotated to GO terms	30,797
Number of genes annotated to KEGG terms	19,737

**Table 2 plants-12-02994-t002:** Primers for amplification of variant sites.

Primer Name ^1^	Primer
GmLox1-snp-F	AGCTTTGGTTGATTTTCTCACAGGT
GmLox1-snp-R	CGGGGATTCCCATGCTTCCG
GmLox1-del-F	CGAGGTAAACATGCGAAGCG
GmLox1-del-R	GCAGCCCATATCTCCAGTCC
GmLox2-snp-F	TGCCACATCCTGCTGGGGA
GmLox2-snp-R	CCGCTGAAGACATCTCAACGGAA
GmLox3-del-F	TACCACCAGGGGCTGTGCTT
GmLox3-del-R	GAAGACACACAAGGAGGACACGC
GmLox1-qPCR-F	ATTGGTTAAATACTCATGCGGC
GmLox1-qPCR-R	CCGAAGACATCTCCACAGAATA
GmLox2-qPCR-F	TCCTGAACAGAGGAGGAGGG
GmLox2-qPCR-R	GTGCCTATGAGTCCCCCAAC
GmLox3-qPCR-F	GCTTGGGGGTCTTCTCCATAG
GmLox3-qPCR-R	GCTGGAGAGACACGGATCG
GmCYP2-qPCR-F	CAAAAACCCTGTCACGCAGT
GmCYP2-qPCR-R	CACTTTCTCTCAAGGGCACCA

^1^ “snp”, “del”, and “qPCR” in primer names indicate that the primer is designed to detect single nucleotide polymorphisms, base deletions, or qRT-PCR, respectively. “F” and “R” represent the forward and reverse primers, respectively.

## Data Availability

The datasets presented in this study can be found in online repositories. The second-generation sequencing data of Dongfudou 3 have been submitted to the NCBI Sequence Read Archive (SRA) (accession number: SRR22692844). All the draft genome information of Dongfudou 3 was stored at http://www.wangsui.net.cn/resource/database/public/plant/Glycine_max/Gmax_DFD3_Survey/Gmax_DFD3_Survey.tar.gz, accessed on 24 May 2023.

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
