# Peer review of "Genome Survey and Chromosome-Level Draft Genome Assembly of Glycine max var. Dongfudou 3: Insights into Genome Characteristics and Protein Deficiencies"

_plants, 2023, doi:10.3390/plants12162994_

Round 1
Reviewer 1 Report
In this manuscript, the authors have sequenced the genome of the soybean cultivar Dongfudou 3 with next-generation sequencing method, and have roughly assembled and analyzed the obtained genome sequence. Subsequently, they tried to uncover why this cultivar lacks the beany flavor based on the sequence data, which may result from the variation of the sequence of the three GmLOX encoding genes. These results are valuable for molecular breeding of the new promising comercial cultivar. But in order to enhance the evidence which can support the conclusion, I think the authors should give more experimental data or discussion as following:
1) Dongfudou 3 lacks the beany flavor, can you give some supplemental quantitative analysis to support this conclusion.
2) The authors predicted that the eliminate of the beany flavor came from the different encoding sequence variations of GmLOX, but actually we still don’t know whether these differences could influence the processes of transcription and/or translation. You can give more analysis to let it make sense.
3) Can you try to give some forward genetic evidence which the mutation or variation of GmLOX could give rise to the difference of beany flavor for soybean, just like you can analyze are there any similar difference in GmLOX locus of cultivars or ecotypes which lack the beany flavor.
Some sentences need to be optimized.
Reviewer 2 Report
It is suggested to do the expression analysis of three genes GmLOX1, GmLOX2, and GmLOX3 in Dongfudou 3 variety, including RNA level and protein level, with variety Zhonghuang 13 as the control.
Moderate editing of English language required
Reviewer 3 Report
The authors investigated the genome survey and chromosome-level draft genome assembly of Glycine max var. Dongfudou 3. Furthermore, the authors gave the insights into genome characteristics and protein deficiencies of Dongfudou 3. In addition, authors also studied the variation of Gmlox gene family between DFD3 and ZH13. This work gives some valuable information for soybean breeding work. However, there are some seriously problems should be revised and considered.
1. The result of characteristics of genome and chromosome-level draft genome assembly of DFD3 is too simple. The author should comprehensive description the genome of DFD3, not just simply taking.
2. The authors should upload the genome and relative data of Dongfudou 3 to the public database such as NCBI. Upload the sequencing data to the public database is the international practices.
3. The authors reported that they establish a simple and fast plant genome survey pipeline method. However, I find nothing results related to this method.

Moderate editing of English language required
Reviewer 4 Report
The paper 'Genome Survey and Chromosome-Level Draft Genome Assembly of Glycine max var. Dongfudou 3: Insights into Genome Characteristics and Protein Deficiencies' aimed to conduct a genomic survey using next-generation sequencing. They determined the genome size, complexity, and characteristics of Dongfudou 3. Additionally, they constructed a chromosome-level draft genome, shedding light on the molecular basis of protein deficiencies in GmLOX1, GmLOX2, and GmLOX3.
The paper is prepared professionally. It includes a well-crafted abstract and an exhaustive introduction that justifies the research undertaken. The introduction points to the deficiencies in the literature on the subject. The aim is clearly defined. Modern analytical methods were used in the research. The discussion of the results is well prepared. The conclusions are well-defined. The illustrative material is appropriate.
In my opinion, the manuscript after corrections, will be suitable for publication in a journal.
Detailed comments:
Abstract:
Do not use abbreviations when use first time.
Introduction - The introduction is enough in my opinion. Introduction needs some minor changes.
Line 30-34 Soybean (Glycine max (Linn.) Merr.), a member of the family Fabaceae, is a crucial crop that provides a major source of proteins and oils for human consumption and livestock feed worldwide. In addition to its nutritional value, soybean also plays a pivotal role in sustainable agriculture by fixing atmospheric nitrogen through a symbiotic relationship with microorganisms
Please give references
I suggest below ones
Novikova, L., Seferova, I., Matvienko, I., Shchedrina, Z., Vishnyakova, M., 2022. Photoperiod and temperature sensitivity in early soybean accessions from the VIR collection in Leningrad Province of the Russian Federation. Turk J Agric For 46 (6):947-954. https://doi.org/10.55730/1300-011X.3055.
Razgonova, M.P.; Zinchenko, Y.N.; Kozak, D.K.; Kuznetsova, V.A.; Zakharenko, A.M.; Ercisli, S.; Golokhvast, K.S. Autofluorescence-Based Investigation of Spatial Distribution of Phenolic Compounds in Soybeans Using Confocal Laser Microscopy and a High-Resolution Mass Spectrometric Approach. Molecules 2022, 27, 8228. https://doi.org/10.3390/molecules27238228
Lavrent'yeva SI, Chernyshuk DK, Martinenko NV, Ivachenko LE, Arsene AL, Ercisli S, Tsatsakis AM, Golokhvast KS, Nawaz MA. Biochemical adaptation of wild and cultivated soybean against toxicity of lead salts. Environ Toxicol Pharmacol. 2020 Oct;79:103429. doi: 10.1016/j.etap.2020.103429.
The paper 'Genome Survey and Chromosome-Level Draft Genome Assembly of Glycine max var. Dongfudou 3: Insights into Genome Characteristics and Protein Deficiencies' aimed to conduct a genomic survey using next-generation sequencing. They determined the genome size, complexity, and characteristics of Dongfudou 3. Additionally, they constructed a chromosome-level draft genome, shedding light on the molecular basis of protein deficiencies in GmLOX1, GmLOX2, and GmLOX3.
The paper is prepared professionally. It includes a well-crafted abstract and an exhaustive introduction that justifies the research undertaken. The introduction points to the deficiencies in the literature on the subject. The aim is clearly defined. Modern analytical methods were used in the research. The discussion of the results is well prepared. The conclusions are well-defined. The illustrative material is appropriate.
In my opinion, the manuscript after corrections, will be suitable for publication in a journal.
Detailed comments:
Abstract:
Do not use abbreviations when use first time.
Introduction - The introduction is enough in my opinion. Introduction needs some minor changes.
Line 30-34 Soybean (Glycine max (Linn.) Merr.), a member of the family Fabaceae, is a crucial crop that provides a major source of proteins and oils for human consumption and livestock feed worldwide. In addition to its nutritional value, soybean also plays a pivotal role in sustainable agriculture by fixing atmospheric nitrogen through a symbiotic relationship with microorganisms
Please give references
I suggest below ones
Novikova, L., Seferova, I., Matvienko, I., Shchedrina, Z., Vishnyakova, M., 2022. Photoperiod and temperature sensitivity in early soybean accessions from the VIR collection in Leningrad Province of the Russian Federation. Turk J Agric For 46 (6):947-954. https://doi.org/10.55730/1300-011X.3055.
Razgonova, M.P.; Zinchenko, Y.N.; Kozak, D.K.; Kuznetsova, V.A.; Zakharenko, A.M.; Ercisli, S.; Golokhvast, K.S. Autofluorescence-Based Investigation of Spatial Distribution of Phenolic Compounds in Soybeans Using Confocal Laser Microscopy and a High-Resolution Mass Spectrometric Approach. Molecules 2022, 27, 8228. https://doi.org/10.3390/molecules27238228
Lavrent'yeva SI, Chernyshuk DK, Martinenko NV, Ivachenko LE, Arsene AL, Ercisli S, Tsatsakis AM, Golokhvast KS, Nawaz MA. Biochemical adaptation of wild and cultivated soybean against toxicity of lead salts. Environ Toxicol Pharmacol. 2020 Oct;79:103429. doi: 10.1016/j.etap.2020.103429.
Round 2
Reviewer 3 Report
The authors have responsed to my comments appropriately.